

# PatCC1: an Efficient Parallel Triangulation Algorithm for Spherical and Planar Grids with Commonality and Parallel Consistency

Haoyu Yang[1], Li Liu[1,2], Cheng Zhang[1,2], Ruizhe Li[1,2], Chao Sun[1], Xinzhu Yu[1], Hao Yu[1], Zhiyuan Zhang[3], Bin Wang[1,2,4]

[1]Ministry of Education Key Laboratory for Earth System Modeling, Department of Earth System Science, Tsinghua University, Beijing, China

[2]Joint Center for Global Change Studies (JCGCS), Beijing, China

[3]Hydro-Meteorological Center of Navy China, Beijing, China

[4]State Key Laboratory of Numerical Modeling for Atmospheric Sciences and Geophysical Fluid Dynamics (LASG), Institute of Atmospheric Physics, Chinese Academy of Sciences, Beijing, China

*Correspondence to:* L. Liu (liuli-cess@tsinghua.edu.cn), Ruizhe Li (liruizhe@tsinghua.edu.cn)

**Abstract.** Graphs are commonly gridded by triangulation; i.e., the generation of a set of triangles for the points of the graph. This technique can also be used in a coupler to improve the commonality of data interpolation between different horizontal model grids. This paper proposes a new parallel triangulation algorithm, PatCC1 (**Pa**rallel **t**riangulation algorithm with **C**ommonality and parallel **C**onsistency, version 1), for spherical and planar grids. Experimental evaluation results demonstrate the efficient parallelization of PatCC1 using a hybrid of MPI and OpenMP. They also show PatCC1 to have greater commonality than existing parallel triangulation algorithms (i.e., it is capable of handling more types of model grids) and that it guarantees parallel consistency (i.e., it achieves exactly the same triangulation result under different parallel settings).

## 1 Introduction

A coupler is a fundamental component or library used in models for Earth system modeling. It handles coupling between component models or even between the internal processes or packages of a component model. A coupler's fundamental functions are data transfer (between different component models, processes, or packages) and data interpolation (between different model grids) (Valcke et al., 2012). Most existing couplers have the capability of remapping coupling fields between different horizontal grids, especially spherical grids. As the horizontal grids of models generally remain unchanged throughout the time integration of a simulation, the data interpolation function of a coupler is generally divided into two stages: the first calculates the remapping weights for a source horizontal grid to a target horizontal grid, and the second uses the same remapping weights to calculate the remapping results at each instance of data interpolation. Most existing couplers can read-in offline remapping weights generated by other software tools such as SCRIP (Jones, 1999),



ESMF (Hill et al., 2004), and YAC (Hanke et al., 2016), while some couplers have the ability of generating online remapping weights.

Commonality can be viewed as a fundamental feature of a coupler. For example, most existing couplers such as OASIS (Redler et al., 2010; Valcke, 2013; Craig et al., 2017), CPL (Craig et al., 2005; Craig et al., 2012), MCT (Larson et al., 2005), and C-Coupler (Liu et al., 2014; Liu et al., 2018) have been used in a range of coupled models. In the past, the longitude–

latitude grid (i.e., a regular grid) was most widely used. However, the rapid development of Earth system modeling has seen various types of new horizontal grid appear, such as the reduced Gaussian grid, tripolar grid, displaced pole grid, cubed-sphere grid, icosahedral grid, Yin-Yang grid, and adaptive mesh, some of which are unstructured. The continuous emergence of new types of horizontal grid introduces a significant challenge to the commonality of couplers, especially the commonality of data interpolation between any two horizontal grids. There are in general two options to address this

challenge: either the new types of horizontal grid are incrementally supported via incremental upgrades of the code of a coupler or remapping software as required, or a common representation is designed and developed for various types of horizontal grid, and then the remapping weights are calculated based on the common grid representation, thus allowing the code of a coupler or remapping software to remain almost unchanged throughout the development of model grids. As the first option will result in the code of a coupler or remapping software become increasingly complicated, the second option is

preferred, provided a common grid representation can be found.

A common grid representation can be achieved by first viewing a grid as a set of independent grid points (only the coordinate values of each point are concerned, while the relationships among grid points—e.g., that one grid point is the neighbor of another—are neglected) and next using one specific gridding method to build relationships among the grid points. Triangulation is a widely used gridding method that generates a set of triangles for independent points in a graph.

Therefore, its use can potentially improve the commonality of data interpolation. In fact, triangulation has already been used by couplers, such as C-Coupler.

Existing triangulation algorithms do not have high time complexity. For example, Delaunay triangulation (Su et al., 1997), which is a widely used triangulation algorithm, has a time complexity of O($N$log$N$) for $N$ points. However, the overhead of triangulation cannot always be neglected, especially as model grids gain increasing numbers of points as the

model resolution increases. Modern high-performance computers equipped with increasing numbers of computing nodes containing increasing numbers of processor cores can dramatically accelerate various applications, including triangulation, that can be efficiently parallelized. MPI (Message Passing Interfaces) is a widely used parallel programming library that can explore the parallelism of processor cores either in the same computing node or among different nodes, while OpenMP is a widely used parallel programming directive that can explore the parallelism of processor cores in the same computing node.

For higher parallel efficiency, many applications (including models for Earth system modeling) have benefited from the hybrid use of both MPI and OpenMP, where MPI generally directs the parallelism among computing nodes and OpenMP controls that of processor cores within the same computing node. Some existing couplers, such as MCT, OASIS3-MCT_3.0 (Craig et al., 2017), and C-Coupler2 (Liu et al., 2018), work as libraries and generally share the parallel setting used by a





component model. When a component model utilizes a hybrid of both MPI and OpenMP for parallelization, a parallel
triangulation algorithm that has been integrated in a coupler will waste the parallelism of processor cores exploited by
OpenMP if the triangulation algorithm only utilizes MPI for parallelization.

Existing couplers such as MCT, CPL6/CPL7, OASIS3-MCT_3.0, and C-Coupler2 can achieve parallel consistency,
which means achieving exactly the same results under different parallel settings. Parallel consistency is important for
debugging parallel implementations. Without it, distinguishing reasonable errors and faults introduced by parallelization is
very difficult. However, the parallelization of triangulation algorithms may damage their consistency. To develop efficient
parallel triangulation algorithms, the entire grid domain is generally decomposed into a set of sub-grid domains, the
triangulation on each sub-grid domain is conducted independently, and the overall result of triangulation is obtained through
merging or stitching the triangles from all sub-grid domains. If the merging or stitching does not force parallel consistency, a
parallel triangulation algorithm may obtain different triangles under different parallel settings. As a result, a coupler may not
be able to guarantee parallel consistency after implementing such a parallel triangulation algorithm.

Therefore, for a triangulation algorithm to be potentially useful in a coupler, it will need to show consistently all three
of the following features: commonality (capable of handling almost every type of model grid), parallel efficiency (efficient
parallelization with a hybrid of MPI and OpenMP), and parallel consistency. There are several parallel triangulation
algorithms that can handle spherical grids (most model grids are spherical grids): e.g., the algorithm proposed by Larrea et al.
(2011) (called the Larrea algorithm hereafter), the algorithm proposed by Jacobsen et al. (2013) (called the Jacobsen
algorithm hereafter), and an improved algorithm based on the Jacobsen algorithm (Prill et al., 2016) (called the Prill
algorithm hereafter). However, none of them simultaneously achieves the three required features (Section 2). With the aim of
achieving these three features, we designed and developed in this work a new parallel triangulation algorithm named PatCC1
(**Pa**rallel **t**riangulation algorithm with **C**ommonality and parallel **C**onsistency, version 1) for spherical and planar grids.
Evaluations using various types and resolutions of model grids and different parallel settings reveal that PatCC1 can handle
various types of model grids, achieve good parallel efficiency, and guarantee parallel consistency.

The remainder of this paper is organized as follows. We briefly introduce related works in Section 2, introduce the
overall design of PatCC1 in Section 3, describe the implementation of PatCC1 in Section 4, evaluate PatCC1 in Section 5,
and briefly summarize this paper and discuss future work in Section 6.

## 2  Related works

This section further introduces the Larrea, Jacobsen, and Prill algorithms in detail.

The Larrea algorithm aims to triangulate global grids. It first uses a 1-D decomposition approach to decompose a global
grid into non-overlapping sub-grid domains of stripes (the boundaries of each sub-grid domain are longitudes), and next
assigns each sub-grid domain to an MPI process (OpenMP is not used in the parallelization) for local triangulation. To obtain
the overall result of triangulation, it collects the local triangles generated by each MPI process and stitches them together

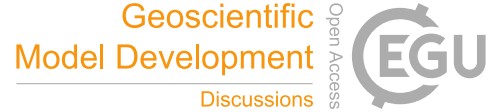

using an incremental triangulation algorithm (Guibas et al., 1985), but without guaranteeing parallel consistency. Therefore, the Larrea algorithm has limitations on commonality, parallel efficiency, and parallel consistency.

The Jacobsen algorithm can triangulate spherical and planar grids. It first decomposes the whole grid domain into partially overlapping circular sub-grid domains, and next instructs each MPI process (OpenMP is not used in the parallelization) to conduct 2-D planar triangulation for a circular sub-grid domain, where the points on a spherical grid are projected onto a plane before the triangulation. To obtain the overall result, it first collects together the local triangles generated by each MPI process, and next scans each triangle, where a triangle is pruned from the overall result if the same triangle already exists. As this algorithm does not check or guarantee parallel consistency, it introduces a risk of overlapping triangles in the overall result. Although it is aimed for use with spherical grids and planar grids, the evaluation in Section 5.2 shows that it is still unable to handle well some types of model grid such as longitude–latitude grids and grids with concave boundaries.

As an upgraded version of the Jacobsen algorithm, the Prill algorithm achieves the following two improvements, but without improving the commonality or the parallel consistency. First, OpenMP is further used in parallelization, which means that parallelization uses a hybrid of MPI and OpenMP. Second, the centers of circular sub-grid domains are determined adaptively, while the circle centers in the Jacobsen algorithm must be specified by the user. The Prill algorithm uses 3-D spherical triangulation implementation rather than 2-D planar triangulation implementation.

## 3 Overall design of PatCC1

The first step of a parallel triangulation algorithm is to decompose the whole grid domain into sub-grid domains. Generally, three questions should be considered in designing a decomposition approach. The first is whether there should be overlapping regions among the sub-grid domains. The Larrea algorithm does not have overlapping regions among the sub-grid domains, so that triangles across the boundaries of sub-grid domains are not obtained through the local triangulation for each sub-grid domain, but are calculated during the last step that obtains the overall triangulation result. We do not prefer such an implementation, as it requires the development of a program that can efficiently calculate in parallel the triangles across boundaries. The second consideration is the choice of the general shape of sub-grid domains. We prefer rectangles rather than the stripes used in the Larrea algorithm and the circles used in the Jacobsen and the Prill algorithms, because the 1-D decomposition corresponding to a petaloid shape will limit the parallelism of a parallel triangulation algorithm, and a circle-based decomposition is disadvantageous in terms of extra overhead. For example, Fig. 1(a) shows a triangle that should be obtained from the correct triangulation of the whole grid domain that is rectangular, and a decomposition of the whole grid domain into four circles. Although these circles are partially overlapping, none of them fully covers the unique triangle in Fig. 1(a). To achieve proper triangulation, these circles should be enlarged accordingly, as in Fig. 1(b), where each circle fully covers the triangle. Figure 1(c) shows a decomposition into four rectangles, each of which also fully covers the triangle. As larger regions of overlap generally mean increased overhead for parallelization, the comparison between Fig.



1(b) and (c) indicates that a circle-based decomposition will introduce higher extra costs than rectangle-based decomposition. The third question is whether it is reasonable to force uniform areas among the sub-grid domains. We prefer to support non-
uniform areas, because the time-complexity as well as the overhead of triangulation is generally determined by the number of grid points, while different sub-grid domains with uniform area may have significantly different numbers of points. In summary, PatCC1 should conduct grid domain decomposition using partially overlapping rectangles of non-uniform area.

The next step after decomposing the whole grid is to triangulate each sub-grid domain separately. Generally, an existing sequential algorithm can be used for this step. Although a spherical grid is on a surface in 3-D space, we prefer 2-D
triangulation algorithms rather than 3-D spherical triangulation algorithms, because the latter generally have relatively complicated implementation and introduce higher computational cost than the former. Experience learned from the Jacobsen algorithm shows that 2-D triangulation can be used after projecting the points in a spherical sub-grid domain onto a plane. However, projection will introduce a challenge to the commonality of parallel triangulation. When there are multiple points corresponding to the same location, projection will implicitly "merge" them into one point, which means only one point is
kept while the other grid points are implicitly pruned. This should not be allowed when multiple points correspond to the same location but have different coordinate values that stand for different grid cells. For example, in a longitude–latitude grid, there are a set of grid points locating at each pole, each of which corresponds to a different grid cell. To overcome this challenge, a step of pre-processing model grids was designed and integrated in the main flowchart of PatCC1.

The next step after local triangulation is to merge the local triangles from all the sub-grid domains together, where the
parallel consistency corresponding to each overlapping region is checked. When an overlapping region fails to pass the checking (which indicates that the corresponding sub-grid domains are not large enough), the corresponding OpenMP threads or MPI processes will enlarge the corresponding sub-grid domains, and then incrementally retriangulate them.

A parallel program generally has limited parallel scalability, which means that lower parallel speedup may be obtained when more processor cores are used. To make the parallel speedup achieved by PatCC1 as high as possible, a computing
resource manager was designed and developed. It first determines the maximum number of processor cores according to the number of points in the grid, and next picks out a set of processor cores that will be used for conducting parallel triangulation. Moreover, it manages the affiliation of each processor core; i.e., which MPI process a processor core belongs to and which OpenMP thread a processor core corresponds to.

Figure 2 shows the main flowchart of PatCC1, which consists of the following main steps:
1)  Pre-process the whole grid;

2)  Initiate the computing resource manager;

3)  Decompose the given model grid into sub-grid domains;

4)  Conduct local triangulation for each sub-grid domain;

5)  Check the parallel consistency: if the parallel consistency is not achieved, go back to the fourth main step to repeat local
triangulation incrementally for the corresponding sub-grid domains after enlarging them;



6) When an overall result of triangulation is required, merge all triangles produced by local triangulations together, after removing repeated triangles.

## 4    Implementation of PatCC1

This section introduces the implementation of PatCC1. In addition to describing each main step in the main flowchart in
Fig. 1, we introduce parallelization with the hybrid of MPI and OpenMP.

### 4.1    Pre-processing of the whole grid

Regarding a spherical grid, PatCC1 takes the longitude and latitude values of each grid point as input, and pre-processes the spherical grid as follows.

1)    The latitude value of each grid point must be between −90° and 90° (or the corresponding radian values). When the
spherical grid is cyclic in the longitude direction, each negative longitude value of grid points will be transformed into the corresponding value between 0° and 360° (or the corresponding radian value). When the spherical grid is acyclic in the longitude direction and the left-most point has a larger longitude value than the right-most point, a transformation will make the longitude values of points monotonically increase from the left side to the right side of the grid. For example, given an acyclic grid with longitude values from 300° to 40°, the longitude values between 300° and 360° will
be transformed to values between −60° and 0°.

2)    If multiple grid points are at the north/south pole and have different longitude values, their latitude value will be changed to a new value that is also the largest/smallest latitude value among all grid points, but is slightly smaller/larger than +90°/−90° (or the corresponding radian values), so that these points will not be the same point after projection. Moreover, a pseudo point at the north/south pole is added to the spherical grid. For example, given a longitude–latitude
grid with a resolution of 1° having 360 grid points at the north and south poles, the latitude values of these points can be transformed to +89.5° and −89.5°, respectively.

Given a regional (not global) spherical grid or a planar grid that is essentially a concave grid (e.g., the grid in Fig. 3(a) that has concave boundaries), as the Delaunay triangulation algorithm cannot handle a concave grid, false triangles will be obtained after triangulation (e.g., the red triangles in Fig. 3(b)). When designing PatCC1, we found that it is difficult to
design a strategy to remove these false triangles. To address this challenge, a set of pseudo grid points on a bounding box of the regional grid is added, which can avoid the generation of false triangles (e.g., the result of triangulation in Fig. 3(c)). After removing the pseudo edges containing pseudo grid points, the result of triangulation can embody the profile of the concave boundaries (e.g., the result in Fig. 3(d)).

### 4.2    Computing resource manager

When using a hybrid of MPI and OpenMP for parallelization, a unique processor core (called a computing resource unit hereafter) is generally associated with a unique thread that belongs to an MPI process. Therefore, the pair *<MPI process ID*,

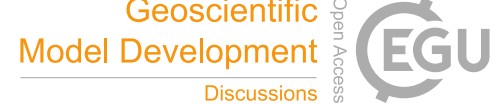

*ID of the thread in the MPI process>* can be used to identify each computing resource unit. The computing resource manager records all computing resource units in an array, where the threads or MPI processes within the same computing node of a high-performance computer correspond to continuous elements in the array. To facilitate the search of computing resource units, the array index is used as the ID of each computing resource unit.

To achieve uniform implementation of parallelization with an MPI and OpenMP hybrid, the computing resource manager provides functionalities of communication between different computing resource units. If two computing resource units are two threads belonging to the same MPI process, the communication between them will be achieved through their shared memory space; otherwise, the communication will be achieved by MPI calls.

As the use of more computing resource units does not necessarily mean faster triangulation, when many computing resource units are available for an insufficiently large number of points in the whole grid, PatCC1 will select a part of the computing resource units for triangulation with the aim of near-optimal parallel performance. To achieve this, the computing resource manager first determines the maximum number of computing resource units according to the number of points in the whole grid and a threshold of the minimum number of points in each sub-grid domain (which can be specified by the user). When the maximum number is smaller than the number of available computing resource units, the computing resource manager will select the same ratio of computing resource units from each computing node. For example, for 1000 available computing resource units, where each computing node includes 20 computing resource units, when the maximum number is 500, 500 computing resource units will be selected, with each computing node contributing 10 computing resource units.

### 4.3 Grid decomposition

The grid decomposition of PatCC1 includes two stages. The first is simultaneously to decompose the whole grid into a set of seamless and non-overlapping sub-grid domains (called kernel sub-grid domains hereafter), assign each kernel sub-grid domain to a computing resource unit, and build a tree for searching kernel sub-grid domains. The second stage produces expanded sub-grid domains through properly enlarging each kernel sub-grid domain, so that at least two expanded sub-grid domains will cover a common boundary between kernel sub-grid domains, and thus parallel consistency can be checked after the triangulation of the expanded sub-grid domains is finished. In the following context, the first and second stages are called kernel decomposition and domain expansion, respectively.

A primary goal of grid decomposition is to achieve balanced triangulation times among sub-grid domains. Although it is difficult or even impossible to achieve absolutely balanced times, we can design a simple heuristic according to the number of points in a sub-grid domain, because the time complexity of triangulation depends on the number of points. The grid decomposition therefore will try to achieve a similar number of points among kernel/expanded sub-grid domains. To facilitate the triangulation for a polar region, the sub-grid domain covering the pole will be circular, while the remaining grid domain that does not cover any pole will be decomposed into a set of rectangles (given a spherical grid, rectangles are defined in longitude–latitude space), as mentioned in Section 3. To avoid narrow rectangles, the grid decomposition should try to achieve a reasonable ratio (e.g., as close to 1 as possible) of the lengths of the edges of each rectangular sub-grid



domain. To avoid the additional work of handling cyclic boundary conditions in triangulation, a cyclic grid domain will be decomposed into a set of (at least two) acyclic rectangular sub-grid domains. Therefore, a global grid will be decomposed into at least four sub-grid domains, even when there are fewer than four computing resource units.

Figure 4 shows the pseudocode of the grid decomposition, where the procedure *decompse_whole_grid* corresponds to kernel decomposition. This procedure takes the whole grid after pre-processing (pseudo points have been added) and the

active computing resource units that have been selected by the computing resource manager as inputs. The free computational capacity of each computing resource unit will be initialized to the number of grid points per computing resource unit (shortened to average point number hereafter), and will be decreased accordingly when a kernel sub-grid domain is assigned to a computing resource unit. A computing resource unit without free computation capacity will no longer be considered in grid decomposition. The procedure *decompse_whole_grid* first generates at most two circular kernel

sub-grid domains with centers at the two poles according to the average point number, whenever the model grid covers either or both poles. Each circular kernel sub-grid domain is assigned to a computing resource unit, and will be inserted into the search tree of kernel sub-grid domains.

The procedure *decompse_whole_grid* next calls the procedure *decompse_subgrid*, which recursively decomposes a given rectangular grid domain for a given set of computing resource units with successive IDs (called a computing resource

set). A cyclic grid domain will be divided into two acyclic sub-grid domains with the same area even when the given computing resource set contains only one computing resource unit. If there is only one computing resource unit, the given rectangular sub-grid domain will be assigned to it. Otherwise, the given computing resource set will be divided into two non-overlapping subsets with balanced total free computational capacity, and two non-overlapping rectangular sub-grid domains will be generated accordingly (their point numbers will be balanced according to the total free computational capacity of the

two computing resource subsets) through cutting the given rectangular grid domain at the long edge. For example, given a rectangular grid domain with 6000 points and a set of five computing resource units (#1–#5) with the same free computational capacity, the two computing resource subsets will include three (#1–#3) and two (#4 and #5) computing resource units, and thus the two rectangular sub-grid domains will contain about 3600 and 2400 points, respectively. Next, the MPI processes that have common computing resource units with the first/second computing resource subset will

recursively decompose the first/second rectangular sub-grid domain, recursively. At each recursion, the newly generated sub-grid domains will be inserted into the domain search tree, as the children of the given grid domain.

The procedure *expand_sub_grid_domain* in Fig. 4 corresponds to the domain expansion stage. It is responsible for the expansion of a given kernel sub-grid domain that has been assigned to the current computing resource unit (a computing resource unit will call this procedure several times when multiple kernel sub-grid domains have been assigned to it). It first

estimates a halo region for expansion based on an expansion rate that can be specified by the user, and then searches the kernel sub-grid domains overlapping with the halo region from the domain search tree. (The search tree will be adaptively updated through a procedure (not shown) similar to the procedure *decompose_subgrid* when it does not include a kernel sub-





grid domain that overlaps with the halo region.) At the same time as generating an expanded sub-grid domain, all neighboring kernel sub-grid domains of the given kernel sub-grid domain will be recorded.

The above design and implementation achieve balanced grid decomposition (balanced numbers of grid points) among the active computing resource units in most cases, and achieve a low time complexity of O($N$) for an MPI process, because the overall domain search tree is almost a binary tree and an MPI process is generally only concerned with a limited number of top-down paths in the tree.

### 4.4  Local triangulation

As introduced in Section 3, we prefer to use a 2-D algorithm in local triangulation. Such an algorithm can directly handle the triangulation of planar grids, while it is necessary to project each sub domain of a spherical grid onto a plane before conducting 2-D triangulation. Similar to the Jacobsen algorithm, the local triangulation of PatCC1 also utilizes stereographic projection, as the Delaunay triangulations on a spherical surface and on its stereographic projection surface are equivalent (Saalfeld, 1999). Our implementation, for a spherical grid, first sets the projection point to the point antipodal to
the center of each spherical sub-grid domain, generates the stereographic projection, and then applies the planar Delaunay triangulation process to the projected points.

For the triangulation process, we developed a divide-and-conquer-based recursive implementation, which in general achieves a time complexity of O($N$log$N$). A recursion of the triangulation implementation is to triangulate the points within a triangular domain. It first finds a point that is near to the center of the triangular domain, and next divides the triangular
domain into two or three smaller triangular domains. Legalization of triangles will be conducted when an illegal triangle is generated (in the Delaunay triangulation, a triangle is illegal if another point is within the circumcircle of the triangle). To avoid frequent memory allocation/deallocation operations that will greatly increase overhead, especially for parallel programs, an optimization of the memory pool is implemented, which efficiently manages the memory usage during triangulation.

There will be multiple legal solutions of Delaunay triangulation in cases having more than three points at the same circle, a situation that is unavoidable or even normal for model grids. When a circle that contains more than three points is in the overlapping region between two expanded sub-grid domains after grid decomposition, local triangulation of the two expanded sub-grid domains may produce different results corresponding to the overlapping region, which means that the triangulation of the whole grid will fail to achieve parallel consistency. A policy was therefore designed and used in the local
triangulation to guarantee parallel consistency: given that the four points of two neighboring triangles (that share two points) are at the same circle, triangulation is legal only when the unique leftmost point or the lower left point (if there are two leftmost points) are not shared by the two triangles (original coordinate values before projection will be used for determining the unique leftmost point or the lower left point). Figure 5 shows an example demonstrating this policy. The triangulation in Fig. 5(a) is illegal, because *P1* is the unique leftmost point but is shared by the two triangles. Fig. 5(b) shows the

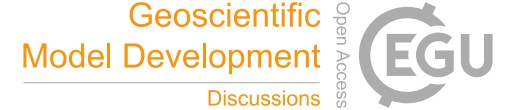

290 corresponding legal triangulation. In Fig. 5(c), both *P1* and *P2* are leftmost points, while *P2* is the lower left point. As *P2* is shared by the two triangles, the triangulation in Fig. 5(c) is illegal. Fig. 5(d) shows the corresponding legal triangulation.

### 4.5 Checking parallel consistency

PatCC1 will examine the parallel consistency of triangulation based on the overlapping regions among the expanded sub-grid domains. When the local triangulations for any pair of overlapping expanded sub-grid domains do not produce

295 exactly the same triangles on the overlapping region, the triangulation for the whole grid fails to achieve parallel consistency. As the local triangulations for a pair of overlapping expanded sub-grid domains are generally conducted separately by different computing resource units, data communication among computing resource units will be required for this step. To reduce the overhead of the data communication, only the triangles across a common boundary between two kernel sub-grid domains are considered, and a checksum corresponding to these triangles will be calculated and used for the checking.

### 300 4.6 Merging all triangles

This main step is optional. It may be unnecessary when the result of triangulation will only be used for generating remapping weights in parallel, because a computing resource unit generally can only consider the sub-grid domains assigned to it in parallel remapping weight generation. This step is necessary when the overall triangulation result will be required. Repeated triangles among different expanded sub-grid domains will be pruned when merging all triangles.

### 305 4.7 Parallelization with an MPI and OpenMP hybrid

To parallelize PatCC1 with a MPI and OpenMP hybrid, we try to parallelize each main step separately, as follows:

1) Pre-processing of the whole model grid. As parallelization of this step with MPI would introduce MPI data communication with a space complexity of $O(N)$, where $N$ is the number of points in the whole model grid, while the time complexity of this step is also $O(N)$, this step is not parallelized with MPI to avoid MPI communication. In other

310 words, each MPI process will pre-process the whole model grid. However, all OpenMP threads in an MPI process will cooperatively finish this step, which means that each OpenMP thread is responsible for pre-processing a part of the points in the whole model grid.

2) Initialization of the computing resource manager. This step will introduce collective communication among the MPI processes. It therefore cannot be accelerated through parallelization, and more MPI processes generally means a higher

315 overhead for this step.

3) Grid decomposition. Similar to the first step, the first stage of this step, which decomposes the whole grid into kernel sub-grid domains, is not parallelized with MPI, while all active OpenMP threads in an MPI process will cooperatively decompose the whole grid. In detail, task-level OpenMP parallelization (corresponding to the OpenMP directive "*#pragma omp task*") is utilized, where each OpenMP task corresponds to a function call of the procedure

320 *decompose_subgrid* if its input sub-grid domain contains enough points (i.e., the point number is larger than a given threshold). In the second stage of this step, each MPI process is responsible for expanding the sub-grid domains that

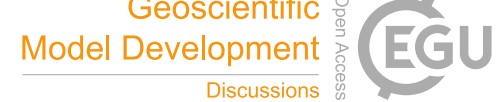

have been assigned to it while task-level OpenMP parallelization is further implemented. Therefore, the second stage has been parallelized with both MPI and OpenMP.

4) Local triangulation. Each computing resource unit is responsible for the local triangulation of the expanded sub-grid
domain assigned to it. Therefore, this step has been parallelized with both MPI and OpenMP.

5) Checking parallel consistency. Parallel consistency is simultaneously checked among different pairs of computing resource units corresponding to different pairs of overlapping expanded sub-grid domains. Therefore, this step has been parallelized with both MPI and OpenMP.

6) Reconducting local triangulation for some sub-grid domains after enlarging them. A computing resource unit is
responsible for its assigned sub-grid domains that fail to pass the parallel consistency check. Therefore, this step has been parallelized with both MPI and OpenMP.

7) Merging all triangles. This step will introduce collective communication among all active computing resource units. It therefore cannot be accelerated through parallelization, and more active computing resource units or more points in the whole grid generally means a higher overhead for this step.

**5   Experimental evaluation**

This section evaluates PatCC1 in terms of commonality, parallel efficiency, and parallel consistency. As the source code of the Jacobsen algorithm is publicly available (https://github.com/douglasjacobsen/MPI-SCVT, last access: 08 Nov 2018), we compared it with PatCC1.

**5.1   Experimental setups**

**5.1.1   Computer platforms**

Two computer platforms are used for evaluation: a shared-memory single-node server and a high-performance computer. The single-node server is equipped with two Intel Xeon E5-2686 18-core CPUs running at 2.3GHz. Simultaneous MultiThreading (SMT) is enabled when using the single-node server, and thus there are 36 physical processor cores and 72 logical process cores. Each computing node of the high-performance computer contains two Intel Xeon E5-2670 v2 10-core
CPUs running at 2.5GHz. SMT is not enabled on the high-performance computer, and there are 20 physical (and thus also logical) processor cores in each computing node. Each computer platform provides enough main memory for evaluation.

Both the Jacobsen algorithm and PatCC1 are compiled with GNU compiler 4.8.5 under the optimization level O3 on either computer platform, and with the same Intel MPI library 3.2.2 on the single-node server and with the same Open MPI library 3.0.1 on the high-performance computer.



### 5.1.2 Model grids

As shown in Table 1, a set of spherical grids from real models are used for evaluation: they are of different types and have different resolutions. Table 2 shows the generation of nine global grids based on three grid types (i.e., longitude–latitude grid, cubed-sphere grid, and randomly generated grid) and three levels of resolution (i.e., coarse, medium, and fine).

### 5.2 Evaluation of commonality and parallel consistency

An algorithm with commonality should successfully triangulate all grids in Tables 1 and 2. Given a whole grid, a successful triangulation should satisfy at least the following criteria:

1) The whole triangulation process finishes normally;

2) Each triangle is a legal Delaunay triangle, and there is no overlapping area between any two triangles;

3) Given that any two grid points do not have the same coordinate values, every grid point is included in at least one
triangle;

4) Each concave boundary (if any) in the original grid is retained after triangulation.

Following the above criteria, PatCC1 successfully triangulates all grids in both tables. Regarding the Jacobsen algorithm, it fails to triangulate all the longitude–latitude grids that cover at least one pole (shown in red in Tables 1 and 2), because the triangulation process will exit abnormally when multiple points are at the same location on the sphere, and there
are a number of points at each pole. It also fails to triangulate the polar grids in Table 1 with concave boundaries. As shown in Fig. 6, the Jacobsen algorithm will generate a number of false triangles above the concave boundaries, whereas PatCC1 does not generate any false triangles. The above results demonstrate that PatCC1 has much greater commonality than the Jacobsen algorithm.

To evaluate parallel consistency, the last main step of PatCC1 is enabled, and all triangles will be written into a binary
data file after sorting them. All grids in both tables are used for this evaluation. At least four parallel settings are used for each grid (with different numbers of MPI processes or different numbers of OpenMP threads). The test results show that for each grid, the binary data files of triangles under all parallel settings are exactly the same. We therefore conclude that PatCC1 achieves parallel consistency.

### 5.3 Evaluation of parallel performance

**5.3.1 Performance on the single-node server**

We first evaluate the parallel performance using all grids in Table 2 on the single-node server. When the total number of processes/threads does not exceed 36, each process/thread will be set to a unique physical core. As the Jacobsen algorithm will use offline grid decomposition information included in two predefined files (one containing a list of region centers for parallelization and the other containing the connectivity of the regions), and three pairs of these files for three parallel
settings (2, 12, and 42 processes) are publicly available (https://github.com/douglasjacobsen/MPI-SCVT, last access: 8 Nov 2018), we use only these three parallel settings to run the Jacobsen algorithm. To compare the Jacobsen algorithm and



PatCC1, we focus only on the time for local triangulation without considering the time for grid decomposition, because the Jacobsen algorithm uses offline grid decomposition information, while PatCC1 calculates grid decomposition information online. According to the test results in Table 3, PatCC1 is faster and achieves higher parallel speedup than the Jacobsen

algorithm in most cases. Moreover, higher parallel speedup is achieved by PatCC1 for finer grid resolution.

To further evaluate the parallel performance of PatCC1 on the single-node server, more parallel settings are used and the time is measured for each main step (except the last step, because it is optional and cannot be parallelized). The test results corresponding to randomly generated grids, cubed-sphere grids, and longitude–latitude grids are shown in Tables 4–6, Tables S1–S3, and Tables S4–S6 (in the supplement), respectively. The results lead to the following observations.

1)   Concurrent running of MPI processes will degrade the performance of the first main step (for pre-processing the whole grid), and more MPI processes generally mean more significant degradation. As this step is memory bandwidth bound and has not been parallelized with MPI, the overall complexity of memory bandwidth requirement is O($MN$), where $M$ is the number of MPI processes and $N$ is the number of grid points. Given $M$ MPI processes, the increment of the run time is generally larger than 1 but much lower than an $M$-fold increase. This is because concurrent running of MPI

processes enables the utilization of more memory bandwidth while  the overall memory bandwidth capacity on a computing node is limited. Regarding OpenMP parallelization, a small parallel speedup (larger than 1) without performance degradation is obtained. This is because the overall complexity of the memory bandwidth requirement remains consistently O($N$), and the concurrent running of OpenMP threads also enables the utilization of more memory bandwidth.

2)   As the second main step (initiating the computing resource manager) will introduce collective communication among MPI processes, the overhead of this step increases with the increment of MPI processes, while the overhead remains almost constant with the increment of OpenMP threads.

3)   Similar to the first main step, the first stage of the third main step (decomposing the whole grid into kernel sub domains) suffers significant degradation when using more MPI processes. Although concurrent running of OpenMP threads can

achieve a faster speed than the concurrent running of MPI processes when the resolution of the grids is medium or fine, more significant performance degradation is also observed when using more OpenMP threads. This is because the overall complexity of the memory bandwidth requirement under OpenMP-only parallelization is O($N\log M$), where $M$ is the number of OpenMP threads, the task-level OpenMP parallelization introduces some extra overhead, and the parallelism exploited is limited. As shown in Table 7, OpenMP parallelization actually accelerates this stage.

4)   As the second stage of the third main step (expanding kernel sub domains) has been parallelized with both MPI and OpenMP, obvious speedup is obtained in concurrent running of MPI processes or OpenMP threads. Compared with MPI parallelization, OpenMP parallelization can avoid redundant grid decomposition among MPI processes (different kernel sub domains assigned to different MPI processes may have the same kernel sub domain as a neighbor), but will introduce the overhead of OpenMP task management and scheduling. As a result, OpenMP parallelization and MPI

parallelization can outperform each other at different grid sizes (i.e., numbers of grid points).



5) As the fourth main step (local triangulation) has been parallelized with both MPI and OpenMP, obvious speedup is obtained in concurrent running of MPI processes or OpenMP threads. Although the same strategy of a computing resource unit only handling the local triangulation of the expanded sub-grid domain that has been assigned to it is employed for both parallelizations, MPI parallelization outperforms OpenMP parallelization in most cases. One possible reason for this is that memory allocation is still necessary in local triangulation after the optimization of the memory pool is implemented, while concurrent MPI processes handle memory allocation generally more efficiently than concurrent threads.

6) Although parallelization with OpenMP or MPI does not achieve obvious parallel speedup for the fifth main step (checking parallel consistency), this step generally takes a small proportion of the overall execution time of parallel triangulation.

7) As SMT can effectively hide the latency from irregular memory access, while frequent irregular memory accesses are introduced by the pointer-based data structures of triangles, SMT provides additional parallel speedup for local triangulation in most cases.

8) Regarding the total execution time, OpenMP-only execution and MPI-only execution can outperform each other at different levels of grid sizes, while hybrid-MPI-OpenMP execution generally achieves a moderate performance between the two.

### 5.3.2 Performance on the high-performance computer

We next evaluate the parallel performance of PatCC1 using the fine grids in Table 2 on the high-performance computer. OpenMP is compared using 1, 5, and 10 threads, and the time for each main step (except the last step) is measured. The test results for the randomly generated grid, cubed-sphere grid, and longitude–latitude grid are shown in Tables 8, S7, and S8 (in the supplement), respectively. Only one computing node is used when there are no more than 20 computing resource units. In addition to the observations discussed in Section 5.3.1, we can make the following observations regarding the increment of computing nodes.

1) The execution time of the first main step remains almost constant with the increment of computing nodes, because the requirement and capacity of the memory bandwidth corresponding to each computing node remain constant.

2) The execution time of the first stage of the third main step increases slightly with the increment of computing nodes, because there will be more recursion levels in grid decomposition when more computing resource units are used.

3) The main step of local triangulation achieves significant parallel speedups. When using 800 processor cores, it achieves more than a 360-fold speedup for all fine grids.

### 5.3.3 Impact of computing resource management

As introduced in Section 4.2, the computing resource manager can adaptively select a part of the computing resource units for triangulation when too many computing resource units are available. To evaluate the benefit of this functionality,



we employ a randomly generated global grid with 2000 points and run PatCC1 on the single-node server under different numbers of MPI processes (MPI only). As shown in Table 9, when this functionality is disabled, after the MPI process number reaches 20, the execution times of local triangulation and the whole PatCC1 algorithm increase with further increases in MPI processes. When this functionality is enabled (the threshold of the minimum number of points in each sub-grid domain is set to 100), after the MPI process number reaches 20, the execution times of both local triangulation and the whole PatCC1 algorithm increase only slightly. (The times for pre-processing the whole grid and initiating the computing resource manager still increase with the increment of MPI processes.)

## 6   Summary and future work

This paper proposes a new parallel triangulation algorithm PatCC1 for spherical and planar grids. Experimental evaluation employing comparison with a state-of-the-art method and using different sets of grids and two computer platforms demonstrates that PatCC1, which has been parallelized with a hybrid of MPI and OpenMP, is an efficient parallel triangulation algorithm with commonality and parallel consistency. Our future work will replace the sequential triangulation algorithm in C-Coupler2 (the latest version of C-Coupler) by PatCC1, so as to develop the next coupler version (C-Coupler3), which is planned to be finished and released before the end of 2021.

When developing the OpenMP parallelization, we preferred to develop coarse-grained rather than fine-grained parallelization, to minimize code modification. Such an OpenMP parallelization achieves obvious parallel speedup for most of the main steps of PatCC1, except the first stage of grid decomposition. We tried to develop a fine-grained OpenMP parallelization for this stage, but without success, because it requires modification of the kernel algorithm, which would thus degrade the performance.

When using a small number of computing resource units, the main step of local triangulation generally takes most of the execution time of the whole PatCC1 algorithm, because the time complexity of each other step is lower. With the increment of computing resource units, the local triangulation is accelerated dramatically, while the non-scalable and low-time-complexity steps (e.g., pre-processing of the whole grid and grid decomposition) gradually become bottlenecks. Our future work will investigate the acceleration of these steps, especially when the grid is extremely large and many computing resource units are used.

The computer platforms used for evaluation in this paper are heterogeneous. To make PatCC1 adapt to a homogeneous computer platform where processor cores have different computing powers, the free computational capacity of each computing resource unit can be initialized according to its computing power.

*Code availability.* The source code of PatCC1 will be publicly available (e.g., through GitLab, GitHub, or another public repository) no later than June 2019.



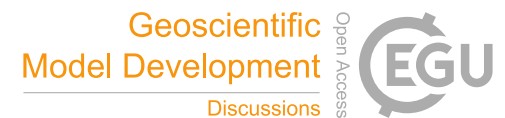

*Author contributions.* HY was responsible for code development, software testing and experimental evaluation of PatCC1,
and co-led paper writing. LL initiated this research, proposed most ideas, supervised HY, and co-led paper writing. RL will
be responsible for code release and technical supports of PatCC1, after HY's graduation. All authors contributed to
improvement of ideas, software testing, experimental evaluation and paper writing.

*Acknowledgements*. This work was jointly supported in part by the National Key Research Project of China (grant nos.
2016YFA0602203 and 2017YFC1501903).

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





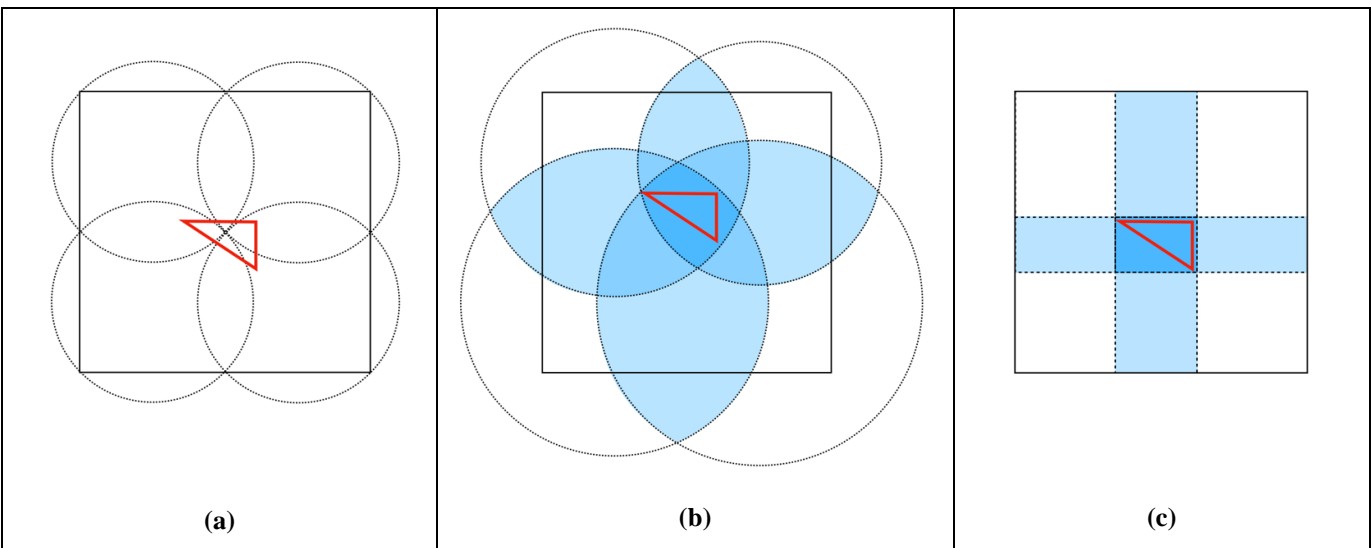


**Figure 1. Examples of overlapping regions under different shapes of sub-grid domains**

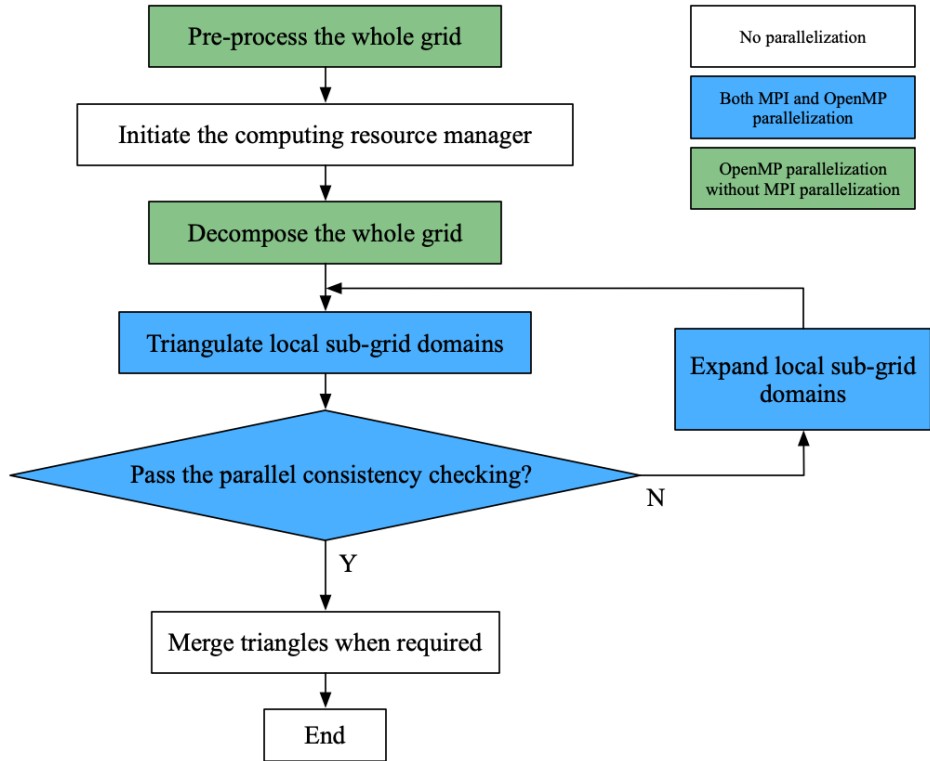


**Figure 2. Main flowchart of PatCC1**

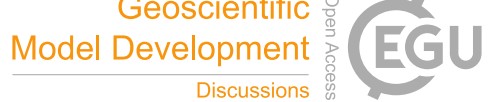



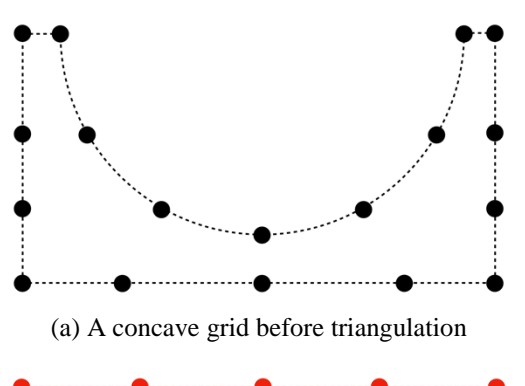

(a) A concave grid before triangulation

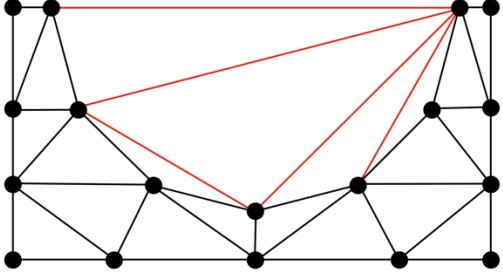

(b) The concave grid after
traditional Delaunay triangulation

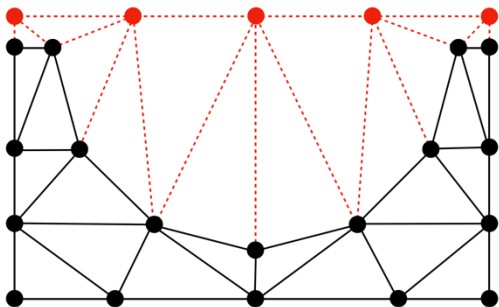

(c) Delaunay triangulation result after adding a set of
pseudo grid points (red) on the bounding box.

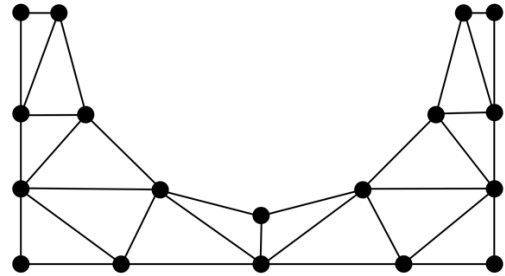

(d) Triangulation result after
pruning the edges with pseudo points in fig. (c)

**Figure 3. Example of adding pseudo grid points to handle the triangulation of a concave grid**


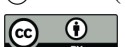



---

**Procedure decompse_whole_grid**

Input: 1) the whole grid *G* after pre-processing; 2) the set of active computing resource units *C*

Output: 1) kernel sub-grid domains, each of which has been assigned to an active computing resource in *C*; 2) search tree of sub-grid

domains

(1) If *G* is a spherical grid and covers the south pole, generate a circular kernel sub domain corresponding to the south pole, assign it to an active computing resource *c1*, and insert it into the search tree; if *c1* does not have free computational capacity for new kernel sub-grid domains, remove *c1* from *C*.

(2) If *G* is a spherical grid and covers the north pole, generate a circular kernel sub domain corresponding to the north pole, assign it to an active computing resource *c2*, and insert it into the search tree; if *c2* does not have free computational capacity for new kernel sub-grid domains, remove *c2* from *C*.

(3) For the remaining sub-grid domain *D*, call *decompose_subgrid*(*D*, *G*)

**Procedure decompose_subgrid**

Input: 1) a sub-grid domain *D*; 2) a set of active computing resource units *C*

Output: 1) kernel sub-grid domains of *D*, each of which has been assigned to an active computing resource in *C*; 2) update of the search

tree of sub-grid domains

(1) If *D* is a cyclic domain and *C* contains only one computing resource unit *c1*, cut *D* into two acyclic sub domains with the same area, assign them to *c1*, insert them into the search tree as the children of *D*, and then return

(2) If *C* contains only one computing resource unit *c1*, assign *D* to *c1* and then return

(3) Divide *C* into two subsets (*C1* and *C2*), which have as equal as possible numbers of computing resource units

(4) Cut *D* into two sub domains (*D1* and *D2*) at the long edge of *D*, according to the total free computational capacity of *C1* and *C2*

(5) Insert *D1* and *D2* into the search tree as the children of *D*

(6) If the current MPI process has common computing resource units with *C1*, call *decompose_subgrid*(*D1*, *C1*)

(7) If the current MPI process has common computing resource units with *C2*, call *decompose_subgrid*(*D2*, *C2*)

**Procedure expand_sub_grid_domain**

Input: 1) a kernel sub-grid domain *D*; 2) a given expansion rate

Output: 1) expanded sub-grid domain of *D*; 2) update of the search tree of sub-grid domains

(1) Estimate a halo region based on the expansion rate

(2) Search the kernel sub-grid domains that overlap with the estimated halo region, generate new kernel sub-grid domains and insert them into the search tree if required.

(3) If the estimated halo region has more points than expected, shrink the halo region gradually

(4) After the halo region is determined, generate the expanded sub-grid domain of *D*, and record the neighborhoods corresponding to *D* in the search tree.

---

**Figure 4.  Pseudocode for grid decomposition**






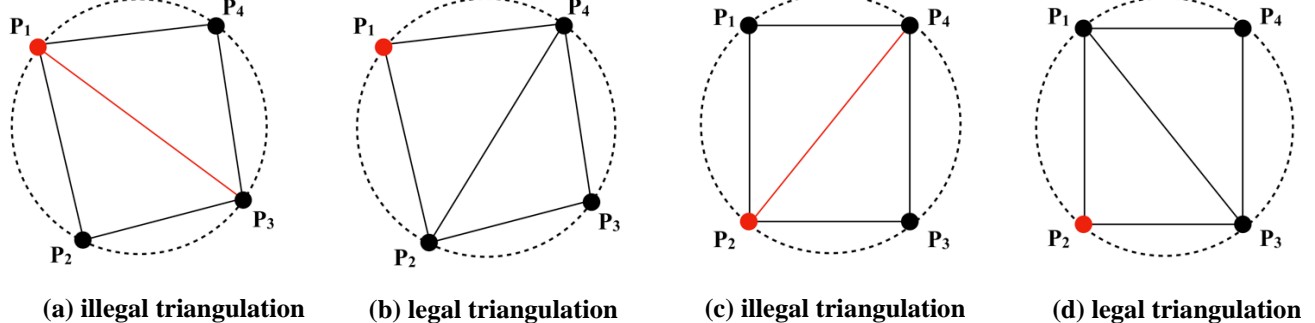

| (a) illegal triangulation | (b) legal triangulation | (c) illegal triangulation | (d) legal triangulation |

**Figure 5. Example demonstrating the policy for guaranteeing a unique triangulation solution. The four points *P1–P4* in each graph (a–d) lie on the circumference of the same circle**





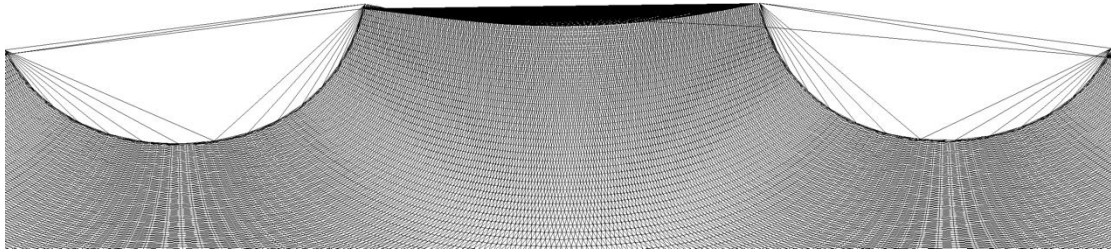

**(a)  Part of triangulation result of the Jacobsen algorithm**

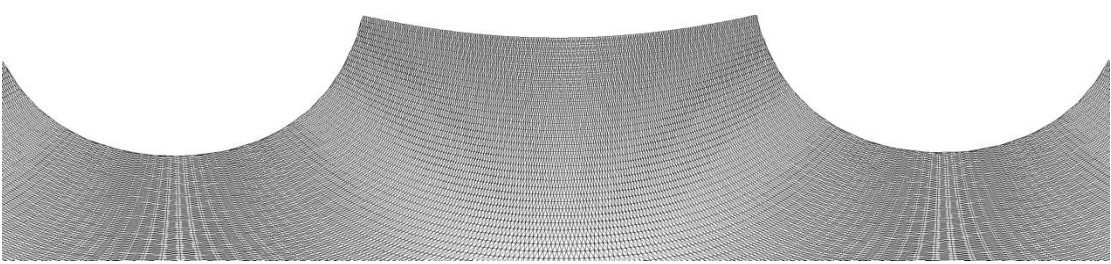

**(b)  Part of triangulation result of PatCC1**

**Figure 6. Triangulation results for the polar grid ar9v4_100920.nc that contains concave boundaries**





**Table 1  Set of spherical grids (from real models) of different types and with different resolutions.**

| Grid type | Name of the grid data file | Number of points | Grid region | Description |
|---|---|---|---|---|
| Polar grid | **ar9v4_100920.nc** | **1684800** | **A north-pole region** | **From the Regional Arctic Climate Model (RACM). This grid has concave boundaries.** |
| | **wr50a_090301.nc** | **56375** | **A north-pole region** | **From the Regional Arctic Climate Model (RACM). This grid has concave boundaries.** |
| Cubed-sphere grid | ne30np4-t2.nc | 48602 | Global region | From the HOMME dynamic core of the atmosphere model CAM |
| | ne60np4_pentagons_100408.nc | 194402 | Global region | From the HOMME dynamic core of the atmosphere model CAM |
| Displaced pole grid | gx3v5_Present_DP_x3.nc | 11600 | Global region without Antarctica | From the ocean model POP |
| | Version_3_of_Greenland_pole_x1_T-grid.nc | 122880 | Global region without Antarctica | From the ocean model POP |
| Longitude–latitude grid | **fv1.9x2.5_050503.nc** | **13824** | **Global region** | **From the finite-volume dynamic core of the atmosphere model CAM** |
| | **licom_eq1x1_degree_Grid.nc** | **70560** | **Global region without Antarctica** | **From the ocean model LICOM** |
| | **licom_gr1x1_degree_Grid.nc** | **61200** | **Global region without Antarctica** | **From the ocean model LICOM** |
| | LICOM_P5_Grid.nc | 242640 | Global region without Antarctica and the north pole | From the ocean model LICOM |
| | **T42_Gaussian_Grid.nc** | **8192** | **Global region** | **From the spectral dynamic core of the atmosphere model CAM** |
| | **T62_Gaussian_Grid.nc** | **18048** | **Global region** | **From the spectral dynamic core of the atmosphere model CAM** |
| | **T85_Gaussian_Grid.nc** | **32768** | **Global region** | **From the spectral dynamic core of the atmosphere model CAM** |
| | **T42_grid.nc** | **8192** | **Global region** | **From the spectral dynamic core of** |





| | | | the atmosphere model CAM |
|---|---|---|---|
| Gamil_2.8_Grid.nc | 7680 | Global region | From the atmosphere grid GAMIL |
| Gamil_1.0_Grid.nc | 64800 | Global region | From the atmosphere grid GAMIL |
| R05_Grid.nc | 259200 | Global region | From a land surface model |



**Table 2. Set of global grids generated in the present study, based on three grid types and three resolution levels.**

| Grid type | Resolution level | Number of points |
|---|---|---|
| **longitude–latitude grid** | **Coarse** | **64800** |
| | **Medium** | **720000** |
| | **Fine** | **6480000** |
| cubed-sphere grid | Coarse | 48602 |
| | Medium | 540002 |
| | Fine | 4860002 |
| randomly generated grid | Coarse | 100000 |
| | Medium | 1000000 |
| | Fine | 10000000 |




**Table 3. Comparison of local triangulation times for the Jacobsen algorithm and PatCC1 under different numbers of**

**total MPI processes.**

| Grid type | Resolution level | Algorithm | Run time (ms) | | | Parallel speedup (2 processes/42 processes) |
|---|---|---|---|---|---|---|
| | | | 2 processes | 12 processes | 42 processes | |
| cubed-sphere grid | Coarse | Jacobsen | 87.3 | 25.2 | 14.9 | 5.87 |
| | | PatCC1 | 111.0 | 37.8 | 14.0 | 7.93 |
| | Medium | Jacobsen | 2,428.8 | 679.0 | 408.0 | 5.95 |
| | | PatCC1 | 1,185.4 | 248.6 | 109.9 | 10.79 |
| | Fine | Jacobsen | 42,466.1 | 16,689.4 | 9,273.0 | 4.58 |
| | | PatCC1 | 12,596.2 | 2,426.0 | 983.7 | 12.80 |
| randomly generated grid | Coarse | Jacobsen | 363.9 | 107.5 | 34.8 | 10.45 |
| | | PatCC1 | 219.6 | 66.1 | 28.8 | 7.61 |
| | Medium | Jacobsen | 10,218.5 | 3,205.4 | 1,902.3 | 5.37 |
| | | PatCC1 | 2,490.5 | 429.2 | 208.6 | 11.94 |
| | Fine | Jacobsen | 392,330.7 | 95,512.4 | 35,366.5 | 11.09 |
| | | PatCC1 | 28,448.2 | 4,672.1 | 2,091.8 | 13.60 |



**Table 4. Run time and parallel speedup of each main step of PatCC1 under different parallel settings, when using the**
**randomly generated grid at the coarse resolution level. "3-1" and "3-2" indicate the first stage (decompose the whole grid into kernel sub-grid domains) and second stage (expand each kernel sub-grid domain) of the third step, respectively. "MPI==OpenMP" indicates that the number of MPI threads and the number of OpenMP threads in each MPI process are equal.**

| Main step ID | Settings of MPI+OpenMP | Run time (ms) under different numbers of computing resource units | | | | Parallel speedup (1 unit/72 units) |
|---|---|---|---|---|---|---|
| | | 1 unit | 6 units | 36 units | 72 units | |
| 1 | MPI only | 0.3 | 1.8 | 1.9 | 3.7 | 0.07 |
| | OpenMP only | 0.3 | 0.2 | 0.1 | 0.1 | 2.03 |
| | MPI==OpenMP | 0.3 | - | 0.6 | - | - |
| 2 | MPI only | 0.030 | 0.076 | 0.388 | 0.998 | 0.03 |
| | OpenMP only | 0.030 | 0.033 | 0.036 | 0.038 | 0.79 |
| | MPI==OpenMP | 0.030 | - | 0.073 | - | - |
| 3-1 | MPI only | 1.3 | 3.0 | 2.5 | 3.7 | 0.34 |
| | OpenMP only | 1.3 | 1.8 | 4.5 | 5.7 | 0.22 |
| | MPI==OpenMP | 1.3 | - | 4.4 | - | - |
| 3-2 | MPI only | 21.0 | 8.9 | 3.5 | 5.0 | 4.18 |
| | OpenMP only | 21.0 | 9.8 | 6.6 | 11.3 | 1.86 |
| | MPI==OpenMP | 21.0 | - | 4.3 | - | - |
| 4 | MPI only | 389.1 | 110.8 | 21.1 | 18.5 | 21.06 |
| | OpenMP only | 389.1 | 118.0 | 50.3 | 63.4 | 6.13 |
| | MPI==OpenMP | 389.1 | - | 28.9 | - | - |
| 5 | MPI only | 0.2 | 0.1 | 0.6 | 0.8 | 0.28 |
| | OpenMP only | 0.2 | 0.2 | 0.5 | 1.3 | 0.16 |
| | MPI==OpenMP | 0.2 | - | 0.5 | - | - |
| Total | MPI only | 411.9 | 124.7 | 30.0 | 32.7 | 12.60 |
| | OpenMP only | 411.9 | 130.0 | 62.1 | 82.0 | 5.03 |
| | MPI==OpenMP | 411.9 | - | 38.8 | - | - |




**Table 5. Run time and parallel speedup of each main step of PatCC1 under different parallel settings, when using the randomly generated grid at the medium resolution level. "3-1" and "3-2" indicate the first stage (decompose the whole grid into kernel sub-grid domains) and second stage (expand each kernel sub-grid domain) of the third step,**
**respectively. "MPI==OpenMP" indicates that the number of MPI threads and the number of OpenMP threads in each MPI process are equal.**

| Main step ID | Settings of MPI+OpenMP | Run time (ms) under different numbers of computing resource units | | | | Parallel speedup (1 unit/72 units) |
|---|---|---|---|---|---|---|
| | | 1 unit | 6 units | 36 units | 72 units | |
| 1 | MPI only | 3.3 | 12.2 | 19.4 | 41.8 | 0.08 |
| | OpenMP only | 3.3 | 1.6 | 1.0 | 1.0 | 3.26 |
| | MPI==OpenMP | 3.3 | - | 5.0 | - | - |
| 2 | MPI only | 0.062 | 0.116 | 0.369 | 1.357 | 0.05 |
| | OpenMP only | 0.062 | 0.070 | 0.072 | 0.070 | 0.89 |
| | MPI==OpenMP | 0.062 | - | 0.105 | - | - |
| 3-1 | MPI only | 10.8 | 26.3 | 32.6 | 65.9 | 0.16 |
| | OpenMP only | 10.8 | 15.5 | 23.5 | 24.1 | 0.45 |
| | MPI==OpenMP | 10.8 | - | 32.6 | - | - |
| 3-2 | MPI only | 184.6 | 54.0 | 32.3 | 44.6 | 4.14 |
| | OpenMP only | 184.6 | 66.0 | 21.1 | 30.5 | 6.06 |
| | MPI==OpenMP | 184.6 | - | 30.1 | - | - |
| 4 | MPI only | 4883.3 | 834.6 | 172.4 | 138.0 | 35.39 |
| | OpenMP only | 4883.3 | 834.6 | 193.1 | 178.8 | 27.32 |
| | MPI==OpenMP | 4883.3 | - | 178.0 | - | - |
| 5 | MPI only | 0.7 | 0.2 | 0.5 | 0.9 | 0.80 |
| | OpenMP only | 0.7 | 0.2 | 0.4 | 1.4 | 0.51 |
| | MPI==OpenMP | 0.7 | - | 0.6 | - | - |
| Total | MPI only | 5082.7 | 927.5 | 257.6 | 292.5 | 17.38 |
| | OpenMP only | 5082.7 | 918.1 | 239.3 | 235.8 | 21.56 |
| | MPI==OpenMP | 5082.7 | - | 246.5 | - | - |



**Table 6. Run time and parallel speedup of each main step of PatCC1 under different parallel settings, when using the randomly generated grid at the fine resolution level. "3-1" and "3-2" indicate the first stage (decompose the whole grid into kernel sub-grid domains) and second stage (expand each kernel sub-grid domain) of the third step, respectively. "MPI==OpenMP" indicates that the number of MPI threads and the number of OpenMP threads in each MPI process are equal.**

| Main step ID | Settings of MPI+OpenMP | Run time (ms) under different numbers of computing resource units | | | | Parallel speedup (1 unit/72 units) |
|---|---|---|---|---|---|---|
| | | 1 unit | 6 units | 36 units | 72 units | |
| 1 | MPI only | 69.3 | 101.2 | 206.3 | 441.9 | 0.16 |
| | OpenMP only | 69.3 | 22.3 | 10.9 | 10.9 | 6.37 |
| | MPI==OpenMP | 69.3 | - | 41.3 | - | - |
| 2 | MPI only | 0.066 | 0.118 | 0.286 | 0.877 | 0.08 |
| | OpenMP only | 0.066 | 0.078 | 0.070 | 0.087 | 0.76 |
| | MPI==OpenMP | 0.066 | - | 0.119 | - | - |
| 3-1 | MPI only | 108.3 | 135.8 | 348.6 | 684.3 | 0.16 |
| | OpenMP only | 108.3 | 152.7 | 293.4 | 319.6 | 0.34 |
| | MPI==OpenMP | 108.3 | - | 224.4 | - | - |
| 3-2 | MPI only | 1772.3 | 372.2 | 317.8 | 448.7 | 3.95 |
| | OpenMP only | 1772.3 | 389.4 | 158.5 | 132.5 | 13.37 |
| | MPI==OpenMP | 1772.3 | - | 263.6 | - | - |
| 4 | MPI only | 58117.3 | 9322.6 | 1662.9 | 1308.8 | 44.41 |
| | OpenMP only | 58117.3 | 9659.6 | 1782.1 | 1381.5 | 42.07 |
| | MPI==OpenMP | 58117.3 | - | 1923.0 | - | - |
| 5 | MPI only | 1.7 | 0.5 | 0.7 | 1.1 | 1.56 |
| | OpenMP only | 1.7 | 1.2 | 0.5 | 1.9 | 0.90 |
| | MPI==OpenMP | 1.7 | - | 1.1 | - | - |
| Total | MPI only | 60069.0 | 9932.5 | 2536.5 | 2885.7 | 20.82 |
| | OpenMP only | 60069.0 | 10225.3 | 2245.6 | 1846.5 | 32.53 |
| | MPI==OpenMP | 60069.0 | - | 2453.5 | - | - |






**Table 7. Run time of step 3-1 with and without OpenMP parallelization when using the randomly generated grid under different resolution levels.**

| Resolution level | Settings of OpenMP | Run time (ms) under different numbers of computing resource units | | | |
|---|---|---|---|---|---|
| | | 1 unit | 6 units | 36 units | 72 units |
| Coarse | With OpenMP | 1.3 | 1.8 | 4.5 | 5.7 |
| | Without OpenMP | 1.2 | 1.6 | 4.0 | 4.7 |
| Medium | With OpenMP | 10.8 | 15.5 | 23.5 | 24.1 |
| | Without OpenMP | 10.6 | 14.3 | 32.4 | 40.0 |
| Fine | With OpenMP | 108.3 | 152.7 | 293.4 | 319.6 |
| | Without OpenMP | 108.3 | 183.6 | 353.5 | 427.7 |




**Table 8. Run time and parallel speedup of each main step of PatCC1 under different parallel settings, when using the randomly generated grid at the fine resolution level. "3-1" and "3-2" indicate the first stage (decompose the whole grid into kernel sub-grid domains) and second stage (expand each kernel sub-grid domain) of the third step, respectively.**

| Main step ID | Settings of MPI+OpenMP | Run time (ms) under different numbers of computing resource units | | | | Parallel speedup (1 unit/800 units) |
|---|---|---|---|---|---|---|
| | | 1 unit | 20 units | 200 units | 800 units | |
| 1 | MPI only | 78.9 | 138.0 | 238.3 | 170.5 | 0.46 |
| | 5 OpenMP threads | 78.9 | 29.9 | 41.4 | 37.9 | 2.08 |
| | 10 OpenMP threads | 78.9 | 19.3 | 18.3 | 17.4 | 4.52 |
| 2 | MPI only | 1.6 | 2.4 | 2.2 | 34.4 | 0.05 |
| | 5 OpenMP threads | 1.6 | 1.6 | 5.8 | 8.1 | 0.20 |
| | 10 OpenMP threads | 1.6 | 1.5 | 2.1 | 0.3 | 5.32 |
| 3-1 | MPI only | 105.8 | 410.6 | 469.4 | 523.3 | 0.20 |
| | 5 OpenMP threads | 105.8 | 178.5 | 202.1 | 181.2 | 0.58 |
| | 10 OpenMP threads | 105.8 | 171.4 | 189.5 | 174.4 | 0.61 |
| 3-2 | MPI only | 1971.7 | 392.9 | 319.7 | 321.9 | 6.13 |
| | 5 OpenMP threads | 1971.7 | 219.4 | 137.8 | 163.3 | 12.07 |
| | 10 OpenMP threads | 1971.7 | 212.6 | 117.7 | 136.5 | 14.44 |
| 4 | MPI only | 58416.1 | 3143.6 | 335.4 | 156.1 | 374.33 |
| | 5 OpenMP threads | 58416.1 | 3216.3 | 341.0 | 154.5 | 378.00 |
| | 10 OpenMP threads | 58416.1 | 3448.3 | 432.0 | 151.4 | 385.87 |
| 5 | MPI only | 2.1 | 33.6 | 74.3 | 136.7 | 0.02 |
| | 5 OpenMP threads | 2.1 | 16.0 | 37.0 | 69.2 | 0.03 |
| | 10 OpenMP threads | 2.1 | 1.9 | 29.9 | 54.8 | 0.04 |
| Total | MPI only | 60576.3 | 4121.2 | 1439.3 | 1342.9 | 45.11 |
| | 5 OpenMP threads | 60576.3 | 3661.6 | 765.1 | 614.3 | 98.61 |
| | 10 OpenMP threads | 60576.3 | 3855.0 | 789.4 | 534.8 | 113.26 |

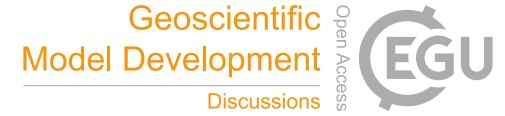


**Table 9. Evaluation of the functionality of adaptively selecting a part of computing resource units for triangulation. A randomly generated global grid with 2000 points is used, and PatCC1 is run on the single-node server under different numbers of computing resource units (MPI only).**

| Adaptive active computing resource units | Main step | Execution time (us) under different number of computing resource units | | | | | |
|---|---|---|---|---|---|---|---|
| | | 1 unit | 10 units | 20 units | 25 units | 36 units | 72 units |
| Disabled | Local triangulation | 23451 | 4572 | 4274 | 4380 | 4676 | 6746 |
| | Whole PatCC1 | 25488 | 5612 | 5891 | 6057 | 6686 | 11613 |
| Enabled | Local triangulation | 23169 | 4572 | 4275 | 4284 | 4279 | 4319 |
| | Whole PatCC1 | 25344 | 5557 | 5917 | 5973 | 6145 | 6606 |