# Peer review of "PatCC1: an Efficient Parallel Triangulation Algorithm for Spherical and Planar Grids with Commonality and Parallel Consistency"

_Geoscientific Model Development, 2018_

## Referee Comment (RC1) · Anonymous Referee #1 · 1 Mar 2019

General Comments

This paper describes the parallel computation of spherical and planar Delaunay triangulations, which can be used by grid point models and related interpolation schemes. The basic idea of the proposed method lies in the efficient combination of well-known building blocks: stereographic projection of subdomains, local triangulation, parallel domain merging and dynamic task scheduling. None of these components is presented in the context of spherical grid generators for the first time. Still, the presented results are interesting and they certainly deserve attention.

Many related subproblems are discussed and handled reasonably, for example the detail of "false triangles", the use of a memory pool, the calculation of finger prints for comparison, and the way how a unique triangulation is chosen from multiple valid Delaunay triangulations.

Specific Comments

- The strength of the paper lies in the discussion of parallel consistency and the hybrid-parallel task scheduling. However, the objective of the overall design should be made more clear: Do the authors aim for a data-parallel algorithm in order to avoid memory bottlenecks? In this case the step (1) in 4.7 would need revision. Otherwise, why should the principal goal be a task parallel algorithm if "most existing couplers can read in offline remapping weights" (l.28)? Furthermore, enforcing a unique triangulation would have no practical use for the calculation of offline remapping weights.
  – Please clarify the design objective.

- Strictly speaking, the paper does not formulate a concise algorithm. The scientific results will be reproducible only after the source code has been published (announced in the manuscript summary).

- The introduction is well written; however, the problem of data interpolation in Earth system modelling is formulated in a rather narrow sense: Vertical remapping, grid staggering, the treatment of over- and undershoots, interpolation of tangent vector fields etc. should be mentioned. All these aspects highly depend on the set of variables and the grid under consideration.

- The proposed algorithm applies to horizontal interpolation of scattered data sets only. Neglecting the grid topology and rebuilding a Delaunay triangulation means

that the algorithm is unsuitable for masking and conservative remapping of finite volume data.

Domain decomposition and pre-processing:

- The decomposition method is actually very similar to classical algorithms like kd-tree half-space subdivision in lon-lat space. The authors should at least expose this similarity and maybe shorten their presentation.

- How do the authors deal with load imbalance due to meridional convergence of a source latitude-longitude grid?

- Points which geometrically coincide at the poles are modified in an elaborate way. For Earth system models, this should not be of practical use, since the source grid points may be topologically different but (should) consistently contain the same value.

- Does the computing resource manager take the faster shared-memory communication into account when decomposing the domain?

Not covered by the manuscript, but of interest:

- A practical parallel algorithm for merging the local triangulations is not presented (e.g. k-way merge)

- Round-off problems, which typically appear (e.g. in the local triangulation step) are not discussed.

- The user-defined expansion rate (l.255) is not explained in detail. How could this rate be determined automatically, ensuring an optimal workload?

Technical Corrections

- l.42: "horizontal grid*s*"
* * *

---

## Referee Comment (RC2) · Anonymous Referee #2 · 15 May 2019

**1    General Comments**

This paper presents a parallel triangulation approach that can be used on spherical and planar grids. The authors establish three main criteria as requirements for a triangulation algorithm: commonality, parallel efficiency, and parallel consistency. The proposed algorithm uses MPI and OpenMP for parallelism of the different steps and their experiments show that it can achieve better scalability, parallel efficiency, and parallel consistency than previous triangulation approaches. The parallel consistency results presented are an improvement over the related algorithms discussed.

[Figure]

**2 Specific Comments**

- The authors provide a clear overview of the goals they have set for their algorithm.

- They describe 3 other parallel algorithms used for Delaunay triangulations and address how PatCC1 differs clearly, which provides the reader an understanding of the intended improvements.

- The paper is comparing PatCC1 (MPI+OpenMP algorithm) with Jacobsen's (MPI only) in Table 3. Please clarify how many OpenMP threads per MPI process were used by PatCC1.

- Obtaining a complete triangulation can be challenging, which is one of the difficulties encountered in the previous algorithms (e.g. stitching, invalid or repeated triangles). The paper would be strengthened by addressing the step or providing a potential parallel approach.

- Given that the algorithm utilizes MPI+OpenMP it would be interesting to scale it to a larger number of nodes to truly observe the impact of internode communication.

- A more detailed description of the implementation of the computing resource manager would be welcomed. The current description favors optimal parallel performance at the cost of leaving resources idle. Perhaps publishing the source code will help clarify this mechanism.

**3 Technical Corrections and Minor revisions**

- The paper states that the source code will be available in June 2019. It would be good for the link to be included in the final version.

- Recommend showing the scaling results using a figure rather than a table.

- Line 36: "new horizontal grid appear" -> "new horizontal grids appear"

- Line 38: "types of horizontal grid" -> "types of horizontal grids"

- Line 40: "types of horizontal grid" -> "types of horizontal grids"

- Line 42: "horizontal grid," -> "horizontal grids,"

- Line 105: "unable to handle well some types..." -> "unable to handle some types of model grids well such as..."

- Line 136: "complicated implementation" -> "complicated implementations"

- Line 146: "the checking" -> "the check"

- Line 165: "Fig. 1" -> "Fig. 2"

- Line 436: "Tables 8, S7, S8" -> what does "S7 and S8" refer to?
* * *

---

## Author Comment (AC1) · 21 May 2019

We thank Reviewer #1 for the comments and suggestions very much. We have modified the manuscript accordingly. In the following, we will reply them one by one.

1. The strength of the paper lies in the discussion of parallel consistency and the hybrid-parallel task scheduling. However, the objective of the overall design should be made more clearly: Do the authors aim for a data-parallel algorithm in order to avoid memory bottlenecks? In this case the step (1) in 4.7 would need revision. Otherwise, why should the principal goal be a task parallel algorithm if "most existing couplers can read in offline remapping weights" (l.28)? Furthermore, enforcing a unique triangulation would have no practical use for the calculation of offline remapping weights. – Please clarify the design objective.

Response: Online remapping weights generation can improve the friendliness of couplers, because users will no longer be forced to manually generate offline remapping weights after changing model grids or resolutions. Some existing couplers such as OASIS and C-Coupler already have the ability of generating online remapping weights. C-Coupler1 and C-Coupler2 have already employed a sequential Delaunay triangulation algorithm for the management of horizontal grids. When cell vertexes of a horizontal grid are not provided, they can be automatically generated from the Voronoi diagram based on the triangulation and further used by non-conservative remapping algorithms (the couplers will force users to provide real cell vertexes of grids involved in conservative remapping).

PatCC1 should be a data-parallel algorithm. To minimize memory usage and synchronizations among computing resource units, we prefer data parallelization for each step of PatCC1, where different computing resource units generally handle different sub-grid domains. Considering that the sub-grid domains to be decomposed dynamically change throughout the main recursive procedure of the grid decomposition (Step 3), we implemented task-level OpenMP parallelization to achieve data parallelization, where all tasks correspond to the same procedure but different sub-grid domains.

The manuscript has been modified accordingly. Please refer to P2L31∼P2L34, P11L350∼P11L354, and P15L487∼P16L490.

2. Strictly speaking, the paper does not formulate a concise algorithm. The scientific results will be reproducible only after the source code has been published (announced in the manuscript summary).

Response: The source code of PatCC1 will be publicly available with the final version of the manuscript.

3. The introduction is well written; however, the problem of data interpolation in Earth system modelling is formulated in a rather narrow sense: Vertical remapping, grid staggering, the treatment of over- and undershoots, interpolation of tangent vector fields etc. should be mentioned. All these aspects highly depend on the set of variables and the grid under consideration.

Response: Other aspects related to data interpolation in Earth system modelling have been mentioned in the revised manuscript. Please refer to P1L24∼P1L25.

4. The proposed algorithm applies to horizontal interpolation of scattered data sets only. Neglecting the grid topology and rebuilding a Delaunay triangulation means that the algorithm is unsuitable for masking and conservative remapping of finite volume data.

Response: C-Coupler1 and C-Coupler2 have already employed a sequential Delaunay triangulation algorithm for the management of horizontal grids. When cell vertexes of a horizontal grid are not provided, they can be automatically generated from the Voronoi diagram based on the triangulation and further used by non-conservative remapping algorithms (conservative remapping algorithms must use the real cell vertexes provided by users). This point has been stated in the revised manuscript (P15L488∼P16L490).

5. The decomposition method is actually very similar to classical algorithms like kdtree half-space subdivision in lon-lat space. The authors should at least expose this similarity and maybe shorten their presentation.

Response: The grid decomposition is similar to k-d tree in longitude-latitude space. This point has been stated in the revised manuscript (P9L265).

6. How do the authors deal with load imbalance due to meridional convergence of a source latitude-longitude grid?

Response: To address this problem, we developed a fast triangulation procedure (its time complexity is O(N)) specific for latitude-longitude grid domains, which will be used

when a polar sub-grid domain has been confirmed as a latitude-longitude grid domain. The manuscript has been modified accordingly. Please refer to P10L298∼P10L303.

7. Points which geometrically coincide at the poles are modiïfied in an elaborate way. For Earth system models, this should not be of practical use, since the source grid points may be topologically different but (should) consistently contain the same value.

Response: As PatCC1 is unable to guarantee that all points at a pole consistently correspond to the same value of each field throughout any model integration, no polar point can be pruned by PatCC1. The manuscript has been modified accordingly. Please refer to P5L42∼P5L146.

8. Does the computing resource manager take the faster shared-memory communication into account when decomposing the domain?

Response: As introduced in the manuscript, if two computing resource units are two threads belonging to the same MPI process, the communication between them will be achieved through their shared memory space; otherwise, the communication will be achieved by MPI calls. In the grid decomposition, shared-memory communication is also used among the OpenMP threads in a process.

9. A practical parallel algorithm for merging the local triangulations is not presented (e.g. k-way merge)

Response: To merge the local triangulations, the root computing resource unit will gather all triangles within or across any boundary of each kernel sub-grid domain from all computing resource units, and then prune repeated triangles (after passing the parallel consistency check, any pair of triangles with overlapping area are the same). The manuscript has been modified accordingly. Please refer to P10L315∼P10L319.

10. Round-off problems, which typically appear (e.g. in the local triangulation step) are not discussed.

Response: The round-off problems have discussed in the revised manuscript

(P16L493∼P16L500).

11. The user-defined expansion rate (l.255) is not explained in detail. How could this rate be determined automatically, ensuring an optimal workload?

Response: The expansion rate has been discussed in the revised manuscript (P8L259∼P9L260, P12L358, P16L501∼P16L505).

12. l.42: "horizontal grids"

Response: This error has been fixed in the revised manuscript.

---

## Author Comment (AC2) · 21 May 2019

We thank Reviewer #2 for the comments and suggestions very much. We have modified the manuscript accordingly. In the following, we will reply them one by one.

1. The paper is comparing PatCC1 (MPI+OpenMP algorithm) with Jacobsen's (MPI only) in Table 3. Please clarify how many OpenMP threads per MPI process were used by PatCC1.

Response: For a fair comparison with Jacobsen's algorithm, PatCC1 only uses one OpenMP thread per MPI process. The manuscript has been modified accordingly.

[Figure]

Please refer to the title of Table 3 (P29).

2. Obtaining a complete triangulation can be challenging, which is one of the difficulties encountered in the previous algorithms (e.g. stitching, invalid or repeated triangles). The paper would be strengthened by addressing the step or providing a potential parallel approach.

Response: To obtain a complete triangulation, the root computing resource unit will gather all triangles within or across any boundary of each kernel sub-grid domain from all active computing resource units, and then prune repeated triangles (after passing the parallel consistency check, any pair of triangles with overlapping area are the same). The manuscript has been modified accordingly. Please refer to P10L315~P10L319.

3. Given that the algorithm utilizes MPI+OpenMP it would be interesting to scale it to a larger number of nodes to truly observe the impact of internode communication

Response: The evaluation in Section 5.3.2 has tried to use more computing nodes. Each computing node contributes 10 processor cores (each computing node has 20 processor cores) when there are 20 computing resource units or more. For example, when there are 800 computing resource units, 80 computing nodes are used. Regarding the internode communication, we can make the following observations from Tables 8, S7, and S8 (in the supplement): 1) The cost of the second step of increases with the number of computing resource units especially the number of processes, because this step introduces collective communications among all computing resource units; 2) The cost of parallel consistency check increases with the increment of computing nodes, and decreases when more OpenMP threads are used under the same number of computing resource units. This is because the parallel consistency check will introduce MPI communications among processes and the overhead of communications generally increases/decreases with the increment/decrement of processes. The manuscript has been revised accordingly. Please refer to P15L456~P15L457, P15L462~P15L463,

P15L468∼P15L471.

4. A more detailed description of the implementation of the computing resource manager would be welcomed. The current description favors optimal parallel performance at the cost of leaving resources idle. Perhaps publishing the source code will help clarify this mechanism.

Response: Parallel efficiency should be a primary goal of a parallel algorithm such as PatCC1. We think that, it will be no additional cost to leave resources idle when more resources used will result in performance degradation. The source code of PatCC1 will be publicly available with the final version of the manuscript.

5. The paper states that the source code will be available in June 2019. It would be good for the link to be included in the iňĄnal version.

Response: The source code of PatCC1 will be publicly available with the final version of the manuscript.

6. Recommend showing the scaling results using a iňĄgure rather than a table

Response: Thanks a lot for this suggestion. In fact, we have tried figures but finally preferred the tables although we know tables are rarely used for showing the scaling results. This is because there will be a lot of figures and it is difficult to observe the comparison of execution time between different steps of PatCC1 at the same parallel setting. We therefore still keep these tables in this revised manuscript and will change them to figures in the next revised version if required.

7. "Tables 8, S7, S8" -> what does "S7 and S8" refer to? Response: "S7 and S8" mean the corresponding to tables in the supplement.

8. Grammatical mistakes Response: All grammatical mistakes listed out have been fixed. Thanks a lot.

---

## Author Response (AR2)

We thank the Editor and Reviewers very much for handling our manuscript.

The code availability section has been modified accordingly, with a DOI link for downloading the code.